# Lunar Procellarum KREEP Terrane (PKT) Stratigraphy and Structure with Depth: Evidence for Significantly Decreased Th Concentrations and Thermal Evolution Consequences

Jingyi Zhang [1,2] , James W. Head [3], Jianzhong Liu [1,*] and Ross W. K. Potter [3,4]

[1] Center for Lunar and Planetary Science, Institute of Geochemistry, Chinese Academy of Sciences, Guiyang 550081, China; zhangjiangyi@mail.gyig.ac.cn
[2] University of Chinese Academy of Sciences, Beijing 100049, China
[3] Department of Earth, Environmental and Planetary Sciences, Brown University, Providence, RI 02912, USA; james_head@brown.edu (J.W.H.); ross_potter@brown.edu (R.W.K.P.)
[4] Institute for Scientific Information, Clarivate, 70 St. Mary Axe, London EC3A 8BE, UK
* Correspondence: liujianzhong@mail.gyig.ac.cn

**Abstract:** Dating from the lunar magma ocean solidification period, the Procellarum KREEP Terrane (PKT) occupies 16% of the surface but has a much higher thorium abundance compared to the rest of the Moon and is thus interpreted to carry 40% of the radioactive elements by volume in the form of an anomalously thick KREEP-rich layer. Subsequent research has focused on the processes responsible for PKT concentration and localization (e.g., degree-1 convection, farside impact basin effects, etc.), and the effect of PKT high-radioactivity localization on lunar thermal evolution (e.g., topography relaxation, mantle heating, late-stage mare basalt generation, etc.). Here we use a stratigraphic approach and new crustal thickness data to probe the nature of the PKT with depth. We find that most PKT characteristics can be explained by sequential impact cratering events that excavated and redistributed to the surface/near-surface a much thinner Th-rich KREEP layer at depth, implying that no anomalous conditions of PKT thickness, radioactive abundances, geodynamics, thermal effects or magma generation are likely to be required as in the previous studies.

**Keywords:** Procellarum KREEP Terrane (PKT); radioactive elements; thorium; thermal anomaly; stratigraphy; structure with depth; Imbrium basin; South Pole–Aitken basin

## 1. Introduction

Understanding the global thermal evolution of the Moon is one of the primary goals of lunar research and exploration. The initial conditions set the stage for subsequent thermal evolution, and a major factor is the nature of early post-accretion history: the formation of a global lunar magma ocean (LMO), the segregation of materials during solidification, the aftermath of solidification (formation of thermal and density instabilities), and most importantly, the mobility and sequestration of radioactive elements during this time. The construction of the initial primary crust [1] and localization of radioactive elements (U, Th, K) in specific crustal and mantle regions can determine the subsequent thermal and geological evolution of the Moon, particularly the generation and longevity of mantle heating and melting, leading to mare volcanism (e.g., [2–6]).

Most lunar evolution models (e.g., [7,8]) predict that the fractional crystallization of a lunar magma ocean will produce a layer of melt enriched in incompatible elements such as K, REE, and P (i.e., KREEP). Several workers (e.g., [9,10]) have proposed that the surficial distribution of Th, which has been measured on a global scale by orbital remote sensing methods for the upper ~30 cm of the Moon [11–16], can be used as a proxy for determining the global distribution of KREEP.

What additional types of data can be used to reconstruct these early elements of crustal formation and lunar thermal evolution? Jolliff et al. (2000) [9] provided an important

interpretive framework by subdividing the lunar surface into three major "terranes" using global Th and FeO [17] distributions and extrapolating the near-surface remote sensing measurements to the lunar crust underlying the surface: (1) Procellarum KREEP Terrane (PKT), (2) Feldspathic Highland Terrane (FHT) and (3) South Pole–Aitken Terrane (SPAT; Figure 1). The FHT (65% of the lunar surface area) was interpreted to represent the upper part of the primary lunar crust, dating to magma ocean solidification. The SPAT (19% of the lunar surface area) was interpreted to represent more mafic lower crustal and perhaps mantle material exposed by the huge oblique South Pole–Aitken basin impact that occurred early in lunar history [18,19] that removed the upper anorthositic crust and redistributed it northward into the surrounding FHT. The PKT (comprising 16% of the lunar surface area) was interpreted to represent an anomalously thick KREEP-rich crust (~60 km thick on the basis of crustal thickness data known at that time [8]) that characterized the NW nearside of the Moon. Globally (Figure 1b), there are obvious high-Th-abundance locations in the PKT (the highest Th value can reach 13 ppm), but almost none in the FHT (<~3 ppm), and only medium-Th concentrations in parts of the SPAT (~4 ppm); the lunar farside generally lacks the high nearside abundances of Th and other KREEP-like elements [20–22]. Jolliff et al. (2000) [9] thus concluded (Figure 1b) that the PKT crust was ~60 km thick, and while representing only 10% of the total lunar crustal volume, contained 40% of the Th in the lunar crust. Clearly, such highly localized concentrations of radioactive elements are predicted to have profound effects on the subsequent thermal and geological evolution of the Moon, leading to "a fundamentally different thermal and igneous evolution within this region compared to other parts of the lunar crust" [9], as explored further by [3–8] and others.

The anomalous concentration of KREEP in the PKT has been cited as a mechanism to explain: (1) the localization of >60% of global mare deposit surface exposures, and the location of virtually all "young" (<3 Ga in age) mare basalts, due to the anomalous heating of the sub-PKT mantle [3]; (2) the lack of distinctive large-scale topographic variations due to viscous relaxation caused by the high geothermal heat flux in the PKT [4]; (3) the apparent absence of magnetic anomalies in the PKT due to the elevated geothermal flux and its effect on delaying the time that magnetic mineral closing temperatures are reached [23,24].

This landmark global terrane classification ([9]; Figure 1) immediately raised some questions about the nature of the structure beneath the PKT and, most importantly, how to account for the PKT and an anomalously thick concentration of KREEP in the Procellarum region. The lateral extent and distribution of the residual magma ocean "KREEP" layer currently remains a matter of debate. Global magma ocean fractional crystallization models generally predict the global presence of a layer of melt enriched in KREEP that solidifies at the base of the crust. This layer may be thinned during final solidification/early aftermath by negative buoyancy that carries dense cumulates and KREEP-rich components into the deeper mantle [1,8,25]. How can the thickness of the KREEP layer increase to the significantly elevated levels (~60 km [8]) proposed for the PKT (Figure 1b; [9]) to concentrate 40% [8] of lunar crustal radioactive elements there?

Models to account for the global asymmetry of KREEP (Figure 1b) and its concentration in the Procellarum KREEP Terrane (Figure 1) include: (1) inhomogeneous differentiation of the magma ocean [26], (2) long-wavelength gravitational instability degree-1 convection during late-stage LMO solidification (mobilizing and transporting farside KREEP to the nearside; [27]), (3) the huge SPA impact basin event facilitating farside KREEP migration to the nearside PKT [28–31] and favoring antipodal PKT KREEP extrusion [32], and (4) the possibility that the Procellarum region is an ancient pre-SPA lunar impact basin, the formation of which may have favored the accumulation of KREEP-rich residual liquid on the nearside [22].

Others have questioned whether the concentration and abundance of Th and the three-dimensional configuration of the PKT described by [9] (Figures 1 and 2) are correct, on the basis of (1) the implied thermal gradient [33], (2) predicted gravity, topography

and electrical conductivity [34], and (3) because large nearside impact basins can possibly explain the configuration without resorting to processes of significantly enhanced KREEP accumulation [21,22,35,36].

Here we use a stratigraphic approach to probe the nature of the PKT with depth. We rely on fundamental stratigraphic principles of unit definition and superposition (e.g., impact crater ejecta excavated from depth and overlying older terrain; lava flows superposed on basin deposits), onlapping (e.g., lava flows embaying older units), and cross-cutting (e.g., a graben or wrinkle ridge cross-cuts several older units) relationships to establish the sequence of events, aided by impact crater size–frequency distributions [37–39].

Using these basic principles, we address the global KREEP asymmetry question from a combined stratigraphic, geological process and geologic history perspective: What is the thickness of the KREEP layer in the PKT, what is the global distribution of KREEP, and how do these findings provide insight into the PKT and early lunar thermal history? We use a synthesis of data from remote sensing, geomorphologic mapping, and impact crater stratigraphic analysis.

## 2. Data and Methods

### 2.1. Thorium Abundances

Thorium abundance from the Lunar Prospector mission [12] is merged with an altimetric shaded relief map [40] and crater depth/diameters from the Lunar Impact Crater Database (2015; https://www.lpi.usra.edu/lunar/surface/, accessed on 12 September 2022). We use impact craters and basins superposed on the ancient lunar crust to document the 3D (lateral and vertical) extent of Th in PKT and SPAT in order to establish the stratigraphy, geometry and potential evolution of the KREEP layer.

### 2.2. Crater Excavation and Sampling Depths

Here, we use relationships between transient cavity depth/diameter [41,42] and crater excavation depth [43,44] to obtain sampling depth estimates (Table 1).

**Table 1.** Impact craters analyzed.

| Name | Coordinates | Thorium Value (ppm) | Diameter (km) | Transient Cavity Diameter [b] (km) | Transient Cavity Depth [b] (km) | Excavation Depth [a] (km) |
|---|---|---|---|---|---|---|
| Diophantus | 27.62°N, 34.3°W | 6.0 | 17.57 | 17.16 | 5.72 | 1.91 |
| Delisle | 29.98°N, 34.68°W | 6.0 | 24.83 | 23.02 | 7.67 | 2.56 |
| Euler | 23.26°N, 29.18°W | 6.0 | 26.03 | 23.96 | 7.99 | 2.66 |
| Lambert | 25.77°N, 20.99°W | 5.8 | 30.12 | 27.13 | 9.04 | 3.01 |
| Kepler * | 8.12°N, 38.01°W | 11.6 | 31 | 26.65 | 8.88 | 2.96 |
| Timocharis * | 26.71°N, 13.10°W | 8.5 | 34.14 | 30.18 | 10.06 | 3.53 |
| Aristarchus * | 27.73°N, 47.49°W | 11.9 | 42 | 34.52 | 11.51 | 3.84 |
| Marian * | 41.60°N, 43.50°W | 11.4 | 39.49 | 34.15 | 11.83 | 3.94 |
| Aristillus * | 33.88°N, 1.21°E | 12.7 | 54.37 | 44.82 | 14.94 | 4.98 |
| Copernicus | 9.62°N, 20.08°W | 7.4 | 96 | 72.71 | 24.24 | 8.08 |
| Plato | 51.62°N, 9.38°W | 5.4 | 100.68 | 75.67 | 25.22 | 8.41 |
| Birkeland * | 30.17°S, 174.01°E | 4.5 | 81.64 | 63.32 | 21.11 | 7.04 |
| Oresme V * | 40.75°S, 165.39°E | 4.9 | 56.1 | 46.03 | 15.34 | 5.11 |

[a] Excavation depth calculated using Melosh (1989) equation; [b] transient cavity diameter and depth from 2015 LPI Impact Crater Database [41,42]; * in Table 1 represent those craters with higher Th content in PKT and SPAT.

This relation between transient cavity diameter and final crater diameter can be written as (Croft, 1985; [41]):

$$D_g \cong D_Q{}^{0.15 \pm 0.04} D_r{}^{0.85 \pm 0.04}$$

where $D_g$ is the transition crater diameter, $D_Q$ is the simple–complex transition diameter, and $D_r$ is the final rim diameter.

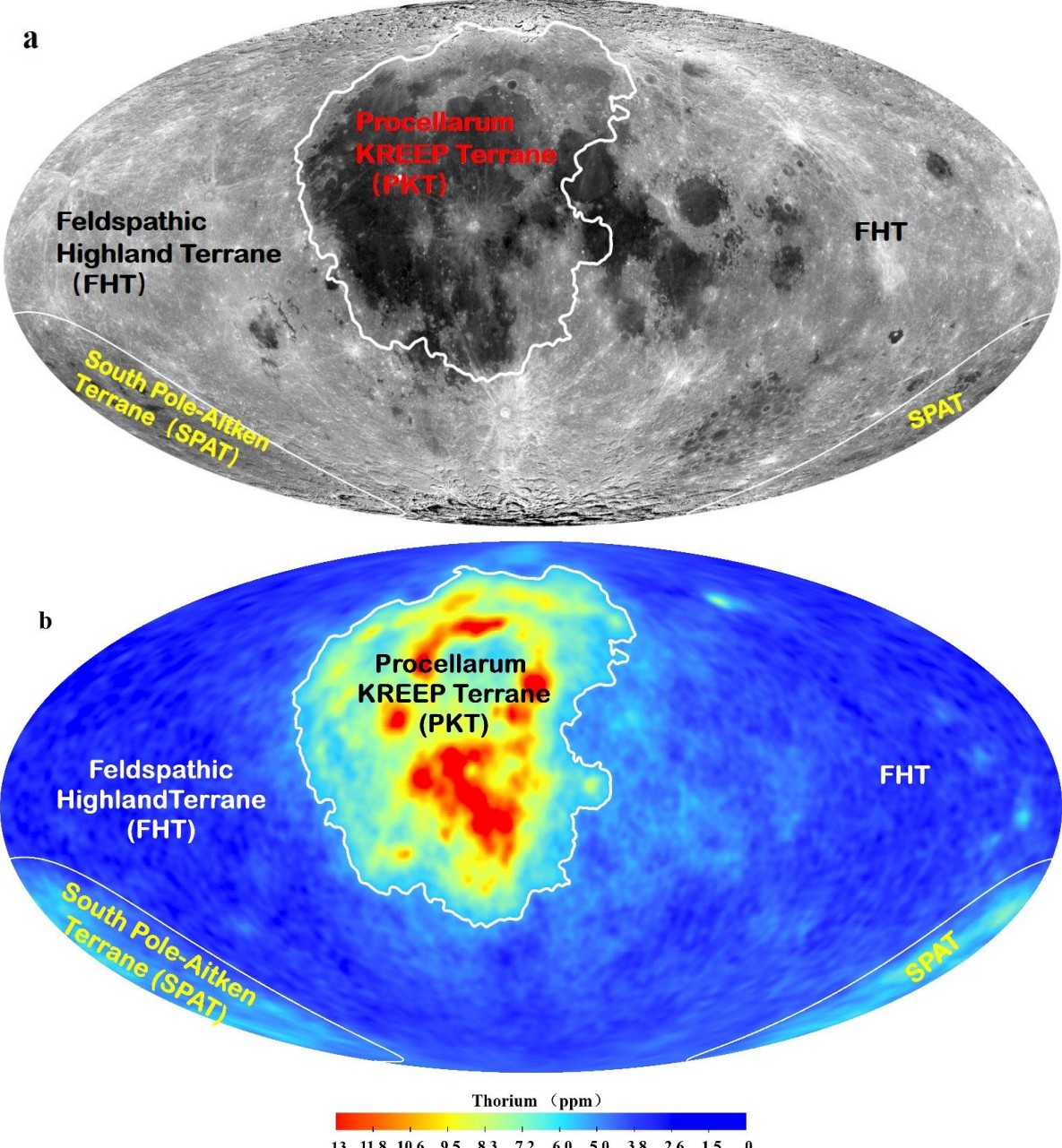

**Figure 1.** Maps showing the three lunar terranes. (**a**) CE-1 global lunar image map (from CE-1 CCD stereo camera DOM-120 m lunar image data). (**b**) Global thorium abundance from Lunar Prospector (LP) gamma-ray spectrometer data (Reprinted/adapted with permission from Ref. [11]. 1998, Lawrence, D.J.) overlain on LOLA hillshade map (Reprinted/adapted with permission from Ref. [40]. 2010, Smith, D.E.). South Pole–Aitken Terrane (SPAT) is marked with white solid line, Procellarum KREEP Terrane (PKT) boundary (red line) is denoted by the contour corresponding to a thorium value of 3 ppm (Reprinted/adapted with permission from Ref. [9]. 2000, Jolliff). Maps are plotted in Mollweide Projection.

Transient cavity depth (km) is equal to 1/3 of transient cavity diameter [42]. For the excavation depth, Melosh [43] provides a conservative rule for the relation between the maximum depth of excavation $H_{exc}$ and the transient crater depth/diameter based on computations and experiments:

$$H_{exc} \cong \frac{1}{3}H_t \cong \frac{1}{10}D_t$$

where $H_{exc}$ is (maximum) excavation depth; $H_t$ is the transient crater depth; $D_t$ is the transient crater diameter.

Potter et al. [44], using numerical models, define the excavation depth by locating the deepest material ejected from the transient crater during excavation, and derive the equation:

$$Z_{ex} = 0.12D_{tc}$$

where $Z_{ex}$ is the excavation depth and $D_{tc}$ is the transient crater diameter for basin-scale impacts.

### 2.3. Recent Reassessment of Global Crustal Thickness and Implications for PKT Radioactive Element Abundance

The definition and assessment of global terrane types [9] utilizes currently known crustal thickness data, with the FHT typically ~70–90 km, the SPAT ~40 km, and the PKT estimated to be ~60 km thick. The GRAIL mission [45] provided global high-resolution gravity data that permitted new, more detailed average crustal thickness estimates and their regional variations [46–48]. GRAIL data revised global crustal thickness values significantly downward, and showed that the crust is even thinner in the PKT region (downward from 60 km to 10–30 km) and the SPAT (downward from 40 km to ~10–20 km). Thus, on the basis of these results alone, the volumetric significance of the PKT radioactive element content, estimated by Jolliff et al. (2000) to be ~40% of the total lunar crust, should be revised downward by ~30–50%, as also suggested by Laneuville et al. (2018) [24].

Based on the above guidelines, we can assess the three-dimensional geometry and stratigraphy of Th concentrations in the PKT, compare this to the SPAT stratigraphy, and outline the hypotheses that seem to best account for Th distribution and characteristics in the PKT and globally.

### 3. Results

### 3.1. Thorium Distribution in the Procellarum KREEP Terrane (PKT)

The Th abundance maps (Figure 2) show that Th levels are very heterogeneous across the PKT. The average Th value for the entire PKT is ~5 ppm, and there is a general trend of decreasing abundance from the center outward to where the PKT grades into the FHT over distances of several hundred kilometers. Within the PKT, there are further variations in Th abundances: (1) areas of younger mare units superposed on the ancient PKT have lower values (4–5 ppm) than other parts of the PKT, but higher values than the mare deposits generally outside the PKT (the easternmost maria have typical values of 1–3 ppm, higher than the FHT, but lower than the maria within the PKT); (2) the non-mare (highland) exposures within the PKT are dominated by the rim of the Imbrium basin and its extensive ejecta deposit, the Fra Mauro Formation (FMF) [37], exposed primarily to the south of the Imbrium basin rim and displaying values of 6–10 ppm; (3) the high Th values exposed around the Imbrium basin rim also show Th variability, with the highest concentrations on the Iridum impact basin rim and ejecta (Figure 2a–c), superposed on the northwest Imbrium basin rim; (4) there are numerous locations of peak Th values (up to 10–12 ppm) within the PKT located at impact craters superposed on both (a) the Imbrium basin rim and ejecta deposits (e.g., Iridum) and (b) younger post-Imbrium-basin mare deposits (e.g., Timocharis and Aristillus; Figure 2b). Other craters superposed on the maria within the PKT show little to no evidence of anomalous Th values (e.g., Lambert, Copernicus; Figure 2b).

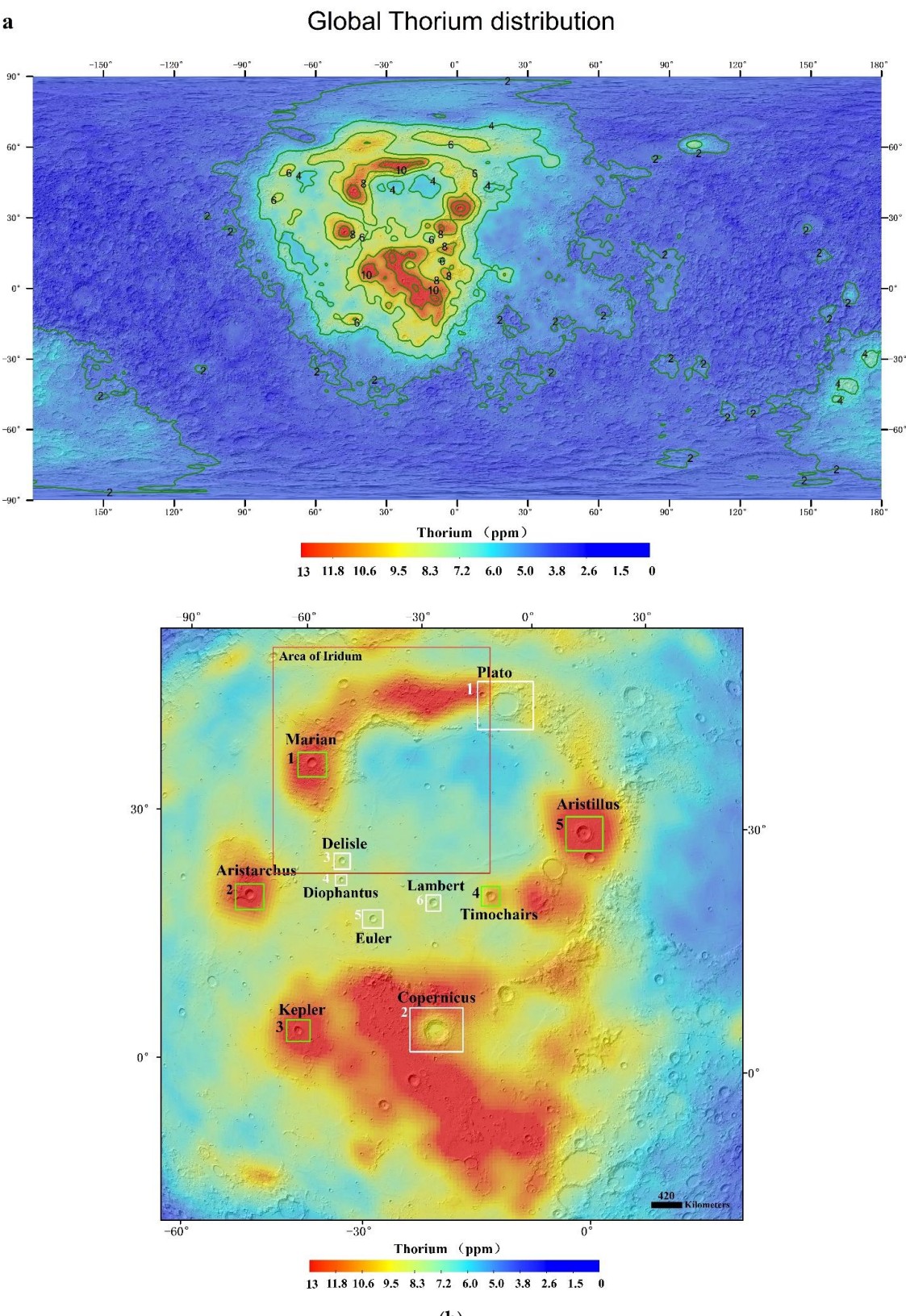

**Figure 2.** *Cont.*

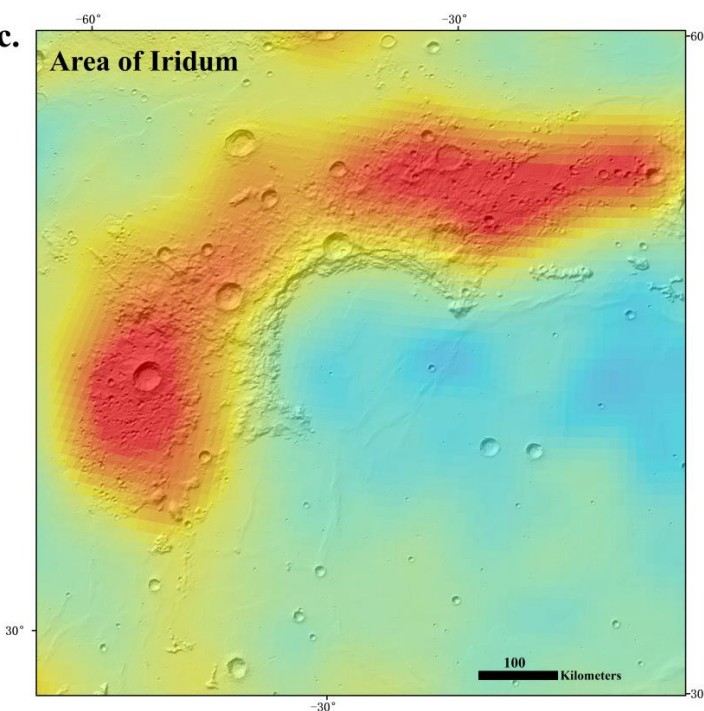

**Figure 2.** (**a**), Global thorium abundance from LP gamma-ray spectrometer (Reprinted/adapted with permission from Ref. [11]. 1998, Lawrence, D.J.) overlain on LOLA hillshade map. Contour lines are labeled in ppm of thorium. (**b**) Thorium abundance in PKT–Imbrium region. Green square boxes and white square boxes show high and low Th concentrations, respectively: green1—Mairan; green2—Aristarchus; green3—Kepler; green4—Timocharis; green5—Aristillus; white1—Plato; white2—Copernicus; white3—Delisle; white4—Diophantus; white5—Euler; white6—Lambert. (**c**) Red dotted square area in (**a**), which represents the Iridum region.

### 3.2. Thorium Distribution in the South Pole–Aitken Terrane (SPAT)

A second concentration of Th occurs in the South Pole–Aitken Terrane (SPAT; Figure 3), where Th levels are higher (~2–9 ppm) than the Farside Highlands Terrane (FHT) values (<~5 ppm), but lower than those in the Procellarum KREEP Terrane (PKT; ~5–13 ppm). Figure 3a shows that the Th contour of SPA and the SPA basin topographic rim overlap significantly (Figure 3a), which means that the relatively high-Th anomaly within the SPAT region and SPA impact basin are closely related [49–52]. The Th distribution map shows that the boundaries of the SPAT are generally more distinct and less gradational than those of the PKT (compare Figures 2a and 3a). In a manner similar to the PKT, Th distribution is also heterogenous within the SPAT (Figures 2a and 3a). Enhanced Th values occur around the northwest region of SPA (Figure 3a) where the two impact craters Birkeland (81.4 km diameter) and Oresme V (56.1 km diameter) show additional peaks of Th abundance (Figure 3b; Table 1).

### 3.3. Thorium Distribution in the Farside Highlands Terrane (FHT)

The analysis of the Th values within the Farside Highlands Terrane (FHT; Figure 2a) shows very low Th abundance over the entire area (<1 ppm), except for one anomalous area, the Compton–Belkovich region on the northwest farside (centered at 61.6°N, 99.5°E). For Compton–Belkovich, the Th values are several ppm (up to ~9 ppm) above the FHT background levels (<1 ppm). This region has been interpreted as the site of post-primary crustal formation, Th-rich, silicic volcanic activity [53]. The Compton–Belkovich Th anomaly is evidence for the presence of KREEP deep in the crust of this region, material that also extruded to the surface near the Humboldtianum impact basin. The lack of additional examples of high-Th features on the farside may be due to the greater farside crustal thickness [54,55],

although some slightly elevated Th anomalies (Figures 1 and 2) are associated with larger impact craters.

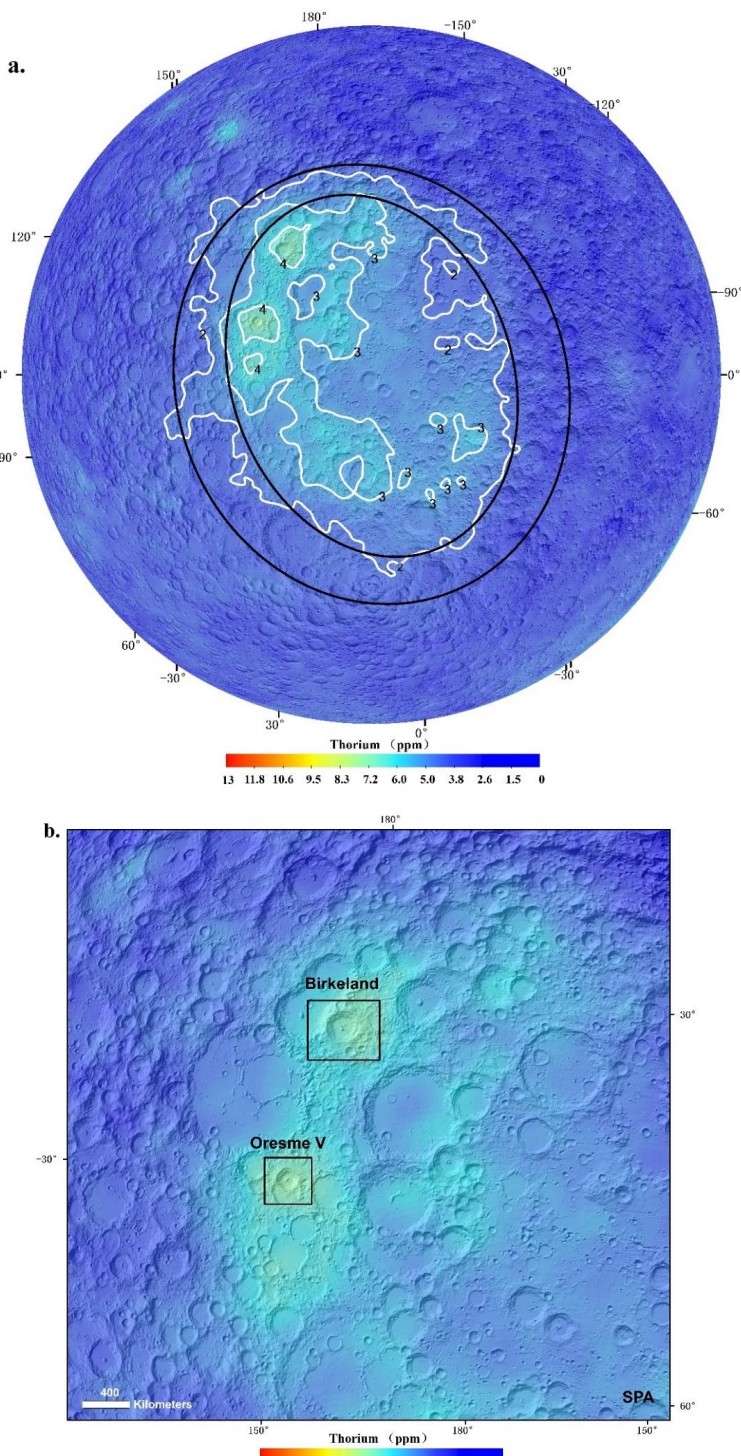

**Figure 3.** Thorium abundance from LP gamma-ray spectrometer (Reprinted/adapted with permission from Ref. [11]. 1998, Lawrence, D.J.) overlain on LOLA hillshade map. (**a**) The white outline represents the Th contour (the black numbers is the Th value with ppm as units) within SPA and corresponds to the 2 ppm Th value contour. Black ovals represent two SPA basin rings (Reprinted/adapted with permission from Ref. [18]. 2009, Garrick-Bethel and Zuber) (**b**) Enlarged map of NW SPA interior showing the craters Birkeland and Oresme V, which show relatively higher Th values (>4 ppm Th) in their ejecta.

### 3.4. Summary

Anomalous Th concentrations associated with crater and impact basin locations (Imbrium, Iridum, SPA) and the related data are presented in Figures 2 and 3, and Tables 1 and 2. These Th maps show that the bulk concentration of Th is clearly inhomogeneous on the lunar surface [13–16]; Th values vary within each terrane, and indeed within each occurrence. In the Procellarum KREEP Terrane (PKT), which is characterized by the highest presence of heat-producing elements (Figure 2a), the high Th concentrations are mostly related to non-mare regions (highland), such as basin rims and ejecta (Figure 2b,c). Other Th anomalies in the PKT are related to several smaller fresh craters (Figure 2b). The South Pole–Aitken Terrane (SPAT) also shows relatively elevated Th compared to the surrounding FHT, but lower than the PKT. There are two high-Th spots within SPA and the SPAT, which are related to two younger impact craters. The entire Farside Highland Terrane (FHT) is a Th-poor terrane at the surface, except for the isolated non-mare silicic volcanism region Compton–Belkovich [53], which displays a distinctive farside high-Th anomaly (Figures 1 and 2a). We now integrate these basic observations and data with our stratigraphic approaches outlined earlier to assess the geologic environments, new crustal thickness data from GRAIL, and excavation depths of superposed craters (Figures 1–3), in order to reconstruct the crustal stratigraphy and assess the implications for the structure and history of the Procellarum KREEP Terrane (PKT).

**Table 2.** Impact basins analyzed.

| Name | Diameter (km) | Transient Cavity Diameter (km) | Transient Cavity Depth (km) | Excavation Depth (km) | |
|---|---|---|---|---|---|
| Iridum | 260 | 134.33 [c] | 44.78 [c] | 16.12 [a] | 13.43 [b] |
| Imbrium | 1160 | 604.23 [c] | 210.41 [c] | 72.51 [a] | 67.1 [b] |
| South Pole–Aitken | 2400 | 840 [d] | 386.85 [d] | 100.8 [a] | 100 [d] |

[a] Excavation depth calculated using the equation of [44]; [b] Excavation depth calculated using Melosh (1989) equation; [c] Transient cavity diameter from 2015 LPI Impact Crater Database ([41,42]); [d] Excavation depth using the equation of [19].

## 4. Discussion

### 4.1. Crustal Stratigraphy Implied by Thorium Anomalies in the SPAT and PKT

The South Pole–Aitken (SPA) basin boundaries (Figures 1 and 3a) are very closely correlated with the topographic margin of the SPA impact basin mapped by Garrick-Bethell and Zuber (2009) [18] and the interior deposits of SPA as documented with Chandrayaan-1 Moon Mineralogy Mapper data [45]. Despite the common opinion that almost all impact cratering events occur at 90-degree incidence angles to the surface, instead, virtually all impacts events are actually oblique to the surface, and a wide variety of workers in the field have documented this fact from observational, theoretical and modeling perspectives [55–58]. The oblique nature of the SPA basin impact, as interpreted from remote sensing data [18], suggests that the basin event removed upper and lower crustal material, ejecting it to the north and leaving more mafic lower crustal material, Th-rich KREEP cumulates, upper mantle, and impact melt deposits exposed within the basin [51,52,59]. The SPAT Th levels (Figure 3) show a low, but slightly elevated concentration within the basin (~2–4 ppm Th) relative to the surrounding Farside Highlands Terrane (FHT; <2 ppm Th) and peaks (4–5 ppm Th) at two post-SPA basin impact craters, Birkeland (Eratosthenian in age; 81.4 km diameter, implying an excavation depth, Z, of ~8 km) and Oresme V (Upper Imbrian; 56.1 km diameter, Z ~5 km excavation depth; Figure 3; Table 1). In the oblique, shallow excavation model [18], one could interpret the stratigraphy of these occurrences (Figure 4a) to indicate the presence of a lowermost subcrustal KREEP layer in the farside SPAT region [50,52], which was brought to the near-surface by the removal of the overlying crust and upper mantle by the oblique SPA basin event, and then excavated and exposed at the surface by the Birkeland and Oresme impact events that penetrated through the SPA basin interior

deposits. In this scenario, the somewhat elevated Thorium values within the SPA basin (~2–5 ppm) are due to (1) the excavation and melting of the extensive KREEP layer in the SPA target area and its inclusion in impact melt and breccias on the basin floor, and (2) the subsequent excavation of the underlying uplifted KREEP layer on the margins of the basin interior by the Birkeland and Oresme impact events (Figure 4a).

In an alternative SPAT scenario, a more vertical impact [51,52] excavates a sub-SPA target KREEP layer and deposits it on the rim together with mantle material ([51]; their Figure 1). In either case, an extensive regional KREEP layer is inferred to exist below the large SPA basin target region [50]. These interpretations thus imply the current presence of a widespread KREEP-rich layer below the lunar farside SPAT at depths in excess of ~5 km (Figure 4a). The lack of significant farside mafic anomalies in the FHT outside of the SPAT, and the fact that the SPA basin is the largest and oldest known basin, leaves open the possibility that this KREEP layer is globally distributed on the lunar farside [51] but is not exposed by subsequent impact events due to thicker farside crust [54] (Figure 3a). Supporting evidence for a more widespread KREEP layer may also lie in the presence of the Compton–Belkovich Th anomaly (Figure 2a), representing Th-rich silicic volcanism derived from melting in a subsurface KREEP-rich layer [53], and in fainter Th anomalies in the FHT at large craters (Figure 2a [56]).

Thorium abundances are much higher (4–12 ppm Th) in the Procellarum KREEP Terrane (PKT) than in surrounding terranes (Figures 1 and 2). The widespread younger mare units within the PKT show much lower Th concentrations (4–6 ppm Th), suggesting that KREEP materials are concentrated in the subsurface below the maria [3,9,12,60]. The highest Th peak anomalies within the PKT (6–12 ppm Th) are mainly related to the Imbrium basin rim and ejecta (Fra Mauro Formation, FMF; Figure 2a,b), and provide insight into the PKT thickness and stratigraphy (Figure 4b). The Imbrium basin, the second-youngest large impact basin, was formed by an oblique impact (projectile approached from the NW at ~30 degrees angle; [61]) that excavated crust and ejected Th-rich KREEP material (Figure 2b). This ejecta was deposited on the basin rim and was spread throughout the lunar highlands as Imbrium basin ejecta—the FMF—particularly to the south (Figure 4b). The post-Imbrium, 259 km Iridum basin lies largely inside the area excavated by Imbrium, and thus penetrates the KREEP-rich layer brought closer to the surface by the Imbrium impact event (Figure 4b, point 4).

Analyzing the high-Th craters in the PKT and their excavation depths (Table 1; Figures 2b and 4b), we found that craters on the highlands (mostly FMF; Mairan, Aristarchus, Kepler; Z ~3.9, 3.8 and 3.0 km, respectively) show the highest Th abundances (11.4–11.9 ppm Th). We interpret these Th concentrations to represent material excavated by the Imbrium impact and distributed only onto the very shallow surface of the highlands (FMF) and then re-excavated by these shallow craters (Figure 4b). The larger craters Copernicus and Plato excavated deeper (Z ~8.1 and 8.4 km, respectively), and thus Th is less abundant at sub-FMF depths (7.4 and 5.4 ppm, respectively; Figure 4b).

Most impact craters in Mare Imbrium and the surrounding maria within the PKT show low Th values (5.8–6.0 ppm) similar to the regional PKT maria (e.g., Lambert, Euler, Delisle, Diophantus; Z ~1.9–3 km; Table 1; Figure 2), suggesting that they have not penetrated through the maria into the underlying KREEP-rich material (Figure 4b, point 6). There are two craters located in Mare Imbrium (Aristillus and Timocharis; Z ~5 and 3.5 km, respectively) that show high Th abundances in their interior and ejecta deposits (8.5 and 12.7 ppm, respectively; Figure 2b) even though the mare is many hundreds of meters thick within the basin [39,54,55,62]. We therefore suggest that both craters penetrated through the mare basalts of Mare Imbrium (Figure 4b, point 5) and excavated underlying Th-rich material associated with the Imbrium basin (impact melt or early KREEP volcanism). This lends support to the model of the Th-rich material being concentrated primarily in Imbrium basin ejecta deposits and the related impact melt and KREEP volcanism, all predating the mare fill. In SPA, observed Th anomalies occur on the outer, more mafic floor of the basin (Figures 3 and 4a), suggesting that their presence is also related to basin formation and is likely indigenous [50–52].

In summary, these data and observations suggest that the current crustal stratigraphy in the Procellarum KREEP Terrane (PKT) and the South Pole–Aitken Terrane (SPAT; Figure 4a,b) differs considerably from that originally proposed by Jolliff et al. (2000) [9]. Specifically, the current PKT subsurface structure (Figure 4b) appears to be layered, with an upper highlands crustal component (<~4–8 km thick) and a lower KREEP-rich layer (at depths >~4–8 km); these are overlain by Th-rich Imbrium–FMF–Iridum basin ejecta deposits (variable thickness, but typically less than ~1–2 km; Figure 4b, points 3,4). Imbrium basin deposits are then flooded and partially covered by mare extrusive volcanic deposits (Figure 4b, point 5); subsequent smaller shallow craters excavate only maria (<~3 km depth; Figure 4b, point 6), but larger craters penetrate through the maria (~3–4 km) and excavate the Th-rich near-surface Imbrium–FMF layers (Figure 4b, point 5). The largest post-basin craters (Copernicus, Plato) penetrate through the FMF–Imbrium deposits into the FHT-like layer that overlies the deeper KREEP-rich layer (Figures 3b and 4b).

**a.**

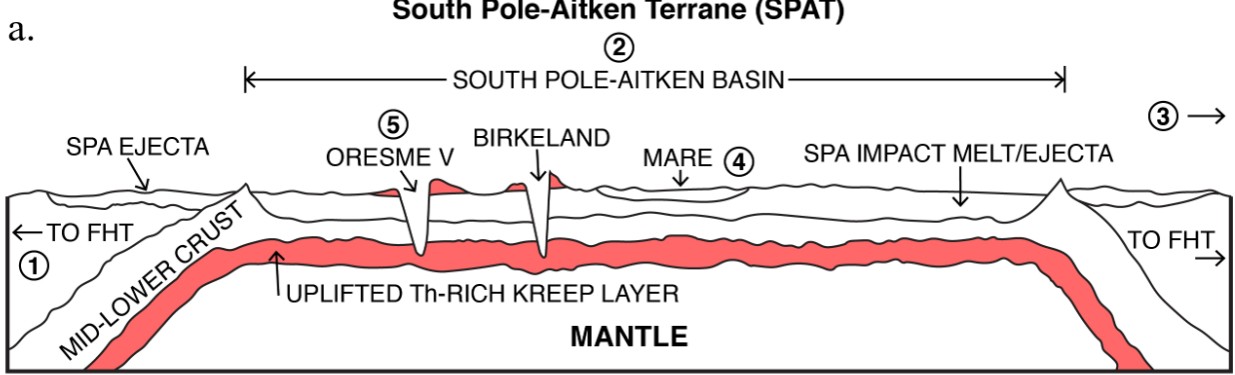

① Initial crust similar to Farside Highland Terrane (FHT).
② Huge, oblique South Pole-Aitken basin impact (~2500 km diameter) strips off upper crust, brings lower crust, KREEP-rich layer to shallower levels, fills SPA basin interior and floor with impact melt deposits and ejecta.
③ Orientale basin event emplaces secondary craters and veneer of ejecta.
④ Mare volcanism floods parts of SPA interior (Ingenii).
⑤ Inside SPA: Larger, deeper Imbrian/Eratosthenian craters Birkeland, Oresme V, penetrate SPA deposits and sample uplifted Th-rich layer at depth.

**b.**

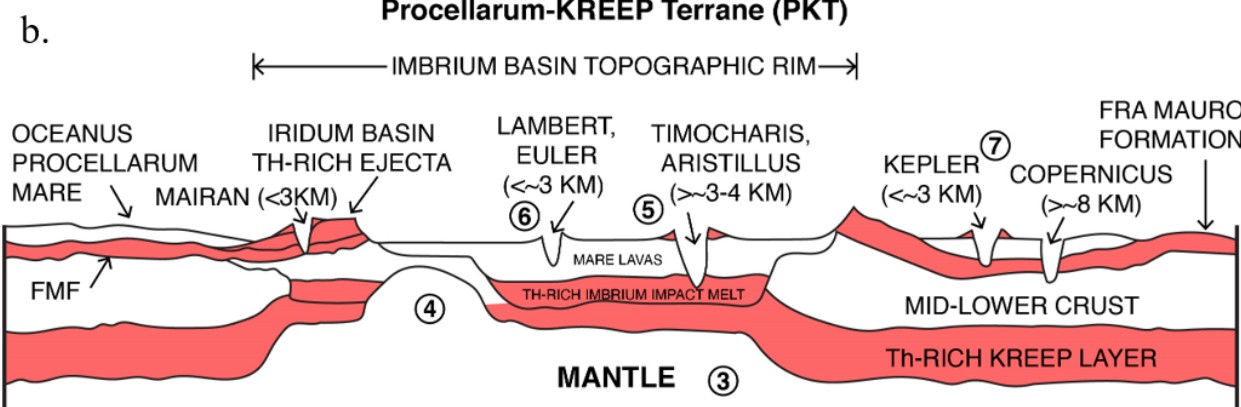

**Figure 4.** Reconstructed crustal cross-sections illustrating the sequence of impact events interpreted from crater/basin age and size in (**a**) South Pole–Aitken Terrane (SPAT; oblique, shallow excavation model) and (**b**) Procellarum KREEP Terrane (PKT). Compare crustal thickness and stratigraphy to original interpretation in Jolliff et al., 2000 [9].

*4.2. Origin of the Procellarum KREEP Terrane*

On the basis of these stratigraphic and geometric relationships as well as new crustal thickness data (Figure 4b) [49,55], we interpret the high Th value distribution in the PKT and SPAT to be related to sequential oblique and nearer-vertical impacts. In order to account for the regionally thin PKT crust [46], an oblique impact forming a Procellarum basin in the PKT region provides one explanation [22], removing much of the upper and middle crust in the area and bringing the residual KREEP-rich layer closer to the surface (Figure 4b). Due to the elevated geothermal gradient of the late-stage LMO period, the basin is predicted to have viscously relaxed, losing much of its topographic and morphologic identity, but retaining its regionally thin crust [63]. At this point in time, the surface of the PKT (Figure 4b) may have looked more like the SPAT of today (Figure 4a): more mafic than the FHT, with Th levels higher than the FHT but lower than the current PKT. The PKT vertical stratigraphy at that time may also have been similar to the oblique, shallow excavation model for the SPAT (Figure 4a) with thinned crust, lower-crustal ferroan anorthosite with modest Th values at the surface, overlying a basal KREEP-rich layer adjacent to the inner basin (Figure 4b).

The formation of a Procellarum basin would have been followed by the formation of the SPA basin [18,32] (Figure 4a), similarly removing much of the upper and middle crust, KREEP-rich layer and upper mantle in the interior, and depositing it on the collapsed basin rim, now in the basin interior [51]. In this scenario, on the nearside, the Imbrium basin impact projectile (approaching from the northwest at ~30-degree angle) would have penetrated the PKT region, through the lower crust and related Procellarum basin deposits, excavating the relatively shallow KREEP layer to form the FMF and related Th-rich ejecta and impact melt deposits (Figure 4b). In this context, the Imbrium basin would have played an analogous role to the smaller craters Birkeland and Oresme V in the SPAT, but on a much larger scale, excavating the deeper KREEP layer and distributing it as Th-rich ejecta onto the lunar surface (Figure 4b).

Following the Imbrium event, the 259 km-diameter Iridum basin formed, depositing Imbrium KREEP on the Iridum rim and the surrounding ejecta deposits, thus producing the currently observed high-Th anomaly (Figures 2b and 4b, point 4). Insight into the PKT vertical crustal stratigraphy is provided by the post-Imbrium/Iridum, ~100 km-diameter Plato crater, which occupies a position on the northern part of the Imbrium basin rim in a more exterior location than that of the larger Iridum basin. The lack of any major Th anomaly associated with Plato suggests that it did not excavate into a KREEP-rich layer, but instead excavated only deep enough (~8.4 km) to reach deeper crustal material above a KREEP layer (Figure 4b). Together, these data suggest the presence of a pre-Imbrium PKT target: an upper SPAT-like ferroan anorthosite layer overlying a lower KREEP-rich layer (Figure 4b). Crater excavation depth data (Table 1) suggest that the KREEP-rich layer was at a depth greater than the Plato excavation of ~8 km.

Evidence supporting this PKT Imbrium basin target stratigraphy (Figure 4b) is provided by other young post-basin craters in the maria and highlands within the PKT [12], including Mairan, Kepler, Aristarchus, Aristillus, Copernicus and Plato (Figure 2a,b and Figure 4b; Table 1). High-Th-concentration regions are near Mairan, Kepler, Aristarchus, and Aristillus (Figures 2b and 4b, point 7). However, not all craters on the Imbrium ejecta show high Th concentrations. Copernicus and Plato (discussed above), located on the FMF and the Imbrium rim, respectively, show relatively lower Th values (Figures 2b and 4b), suggesting that the high-Th KREEP terrane does not exist directly below surficial FMF ejecta deposits in these areas. Craters within Mare Imbrium generally show lower Th abundances than those on the highlands (suggesting that they excavated maria, not Th-rich FMF or crust; Figures 2b and 4b, point 6), with two exceptions, Timocharis and Aristillus (Figures 2b and 4b, point 5). These two craters appear to excavate through Th-poor maria into Th-rich Imbrium basin deposits (FMF etc.; Figure 4b). Mare basalt eruptions subsequent to the Imbrium and Iridum basin impact events buried the KREEP-rich deposits under hundreds of meters to kilometers of lava [39,60–62]; the young Procellarum basalts thus required no anomalous KREEP concentration in their generation, ascent and em-

placement [64–66]. Within SPA, similar impacts (Birkeland and Oresme V) excavated Th-rich material [50] within ejecta deposits (Figure 4a, point 5) to create the locally high Th anomalies.

In summary, this new interpretive scenario (Figure 4) attributes the origin of the PKT and SPAT Th anomalies to sequential impact processes, requiring no additional processes (such as inhomogeneous magma ocean differentiation or heterogeneous degree-1 convection) to concentrate KREEP in these terranes at depth.

## 5. Conclusions

The current paradigm regarding the ancient lunar crust is that it consists of three main terranes mapped on the basis of global Th and FeO data (PKT, SPAT and FHT) [9,10]. The NNW nearside PKT, covering only 16% of the surface, has been interpreted to contain ~40% of the total crustal radioactivity, a concentration called on to explain the subdued topography, the location of >60% of the maria by area, and the anomalous longevity of the mare eruptions in the PKT (mare basalts <~3 Ga).

We used the global Th data to examine Th values associated with impact craters and basins, and new GRAIL data on crustal thickness, to assess the three-dimensional structure of the three terranes, and found that the PKT crust is less than half the thickness previously thought, and that the observed high Th values can be explained by a series of impacts penetrating into a low-Th layered crustal stratigraphy and into a basal KREEP-rich layer, thus redistributing high-Th material as surficial deposits (Figure 4). This interpretation does not require any special mechanisms to concentrate global KREEP-rich material in the PKT, is consistent with observations of the SPAT, and can be tested with an analysis of PKT mare basalt and regolith samples. For example, an analysis of the returned Change 5 samples, collected from a young mare basalt unit in the central part of the Procellarum KREEP Terrane [67], showed a young eruption age [65,68] but little evidence for being Th-rich or having been derived from a KREEP-rich mantle source [66]. Without a doubt, the PKT has a profound influence on the lunar thermal evolution; nevertheless, based on a thinner crust and lower content of radioactive elements in the PKT than previously predicted (still the highest level compared to other regions), the PKT may not seem as extremely significant as has been inferred in the past. Some outstanding remaining questions include: Why is the heat flow at the Apollo 15 site high compared to the Apollo 17 site? Could the history outlined here account for the existence of low magnetism in the PKT area [23]? Could the early crustal and lithospheric thinning in this scenario account for the apparently larger impact basins on the lunar nearside [47]?

**Author Contributions:** All authors made significant contributions to the work. J.Z.: methodology, investigation, writing—original draft preparation; J.W.H.: conceptualization, formal analysis, invention, writing—review & editing; J.L.: conceptualization, investigation, writing—review & editing, funding acquisition; R.W.K.P.: review & editing. All authors have read and agreed to the published version of the manuscript.

**Funding:** This work was supported by National Key Research and Development Program of China (Grant No. 2022YFF0503100) the Strategic Priority Research Program of Chinese Academy of Sciences, (Grant No. XDB 41000000), National Natural Science Foundation of China (Grant No. 42202264). J.W.H. was supported by non-grant research in conjunction with his teaching and research position at Brown University.

**Data Availability Statement:** Thorium concentration data reproduced the figures in this work can be accessed through the Planetary Data System Geosciences Node (https://pds-geosciences.wustl.edu/missions/lunarp/reduced_special.html, accessed on 16 December 2018). Some craters diameter data used in this work can be accessed via 2015 lunar crater database (https://www.lpi.usra.edu/lunar/surface/, accessed on 12 September 2022).

**Acknowledgments:** We gratefully acknowledge very helpful discussions with E. M. Parmentier, Malcolm J. Rutherford, Alberto Saal, Stephen Parman and Paul C Hess. We Also acknowledge GRAS (https://moon.bao.ac.cn, accessed on 12 December 2018) for the Chinese mission data. We also thank the Lunar Prospector and Clementine teams for providing the data used in this work.

**Conflicts of Interest:** The authors declare no conflict of interest.

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
