# Peer review of "Lunar Procellarum KREEP Terrane (PKT) Stratigraphy and Structure with Depth: Evidence for Significantly Decreased Th Concentrations and Thermal Evolution Consequences"

_remotesensing, doi:10.3390/rs15071861_

Round 1
Reviewer 1 Report
This paper proposes the use of stratigraphic methods to probe the nature of PKT with depth. The authors' results found that most of the PKT characteristics can be explained by sequential impact cratering events to excavate and redistributed thinner KREEP layers. After going through the whole paper, generally speaking, the paper is well organized and the research methods are appropriate, which makes it a qualified scientific paper. However, there are still existing areas that need to be improved. Detailed comments are as follows.
1. The authors need to explain why the relationship between PKT thickness, radioactive abundances, geodynamics, and thermal effects or magmatogenesis is not relevant.
2. In the introduction section, the authors need to clarify why the stratigraphic method is used in this paper. Why is the stratigraphic method better than previous studies? What are the advantages? What are the shortcomings of the current studies in this field? How the authors have improved this study, which also needs to be clarified.
3. Figure 1, which needs to indicate the source of the image data used.
4. What is the orange text in Table 1? Is it a writing error?
5. In Figure 2, the latitude and longitude characters are too small and need to be enlarged.
6. In Figure 3, some numbers are not recognizable.
7. What the authors said about the nature of the PKT with depth, and how is the depth reflected in it, needs to be clear.
Author Response
Thank you very much for the reviewer's comment.
We have answered these comments points by points and summarise a cover letter.
Please see the attachment.

Reviewer 2 Report
General aspects
The work is about to give some insight into the terrain composition and arrangement on the Moo, using thorium concentration at crater exhumed locations. The topic is interesting, new results are presented and being the interest of the readers. The methods are good, the structure and the language are very good. Some minor comments are presented below, suggesting minor revise.
At several times the “oblique impacts” are mentioned, please indicate why it is reasonable to be “oblique”.
Specific aspects
36
“ volcanism (e.g., [2-5]).”
suggest to cite here also the overview of various vulcanic features on the Moon in the recent work: https://ui.adsabs.harvard.edu/abs/2015epl..book.....H/abstract
64
“all young (<3 Ga in age)”
suggest to put the young to quotation mark
109
“Lunar Prospector mission [11] are merged”
would be „is” better than „are”?
Figure 2 b and c
scale bar is needed
Figure 3 b
scale bar also needed in the right panel
242
“and desposts it mon the rimetogether”
afraid several type errors emerged here, please rewrite
375
“no additional processes to concentrate”
could you briefly list in brackets, what would be these additional processes?
Author Response
Thank you very much for the reviewer's comments.
We make a cover letter to revise these comments point by point.
Please see the attachment

Round 2
Reviewer 1 Report
I don't have any more questions. Good luck to the authors.